# Black Grouse Hissing Calls Show Geographic Variability across the Distribution Area

**DOI:** 10.3390/ani13111844

**Published:** 2023-06-01

**Authors:** Lucie Hambálková, Richard Policht, Jan Cukor, Vlastimil Hart, Richard Ševčík

**Affiliations:** 1Department of Game Management and Wildlife Biology, Faculty of Forestry, Czech University of Life Sciences, Kamýcká 129, Suchdol, 165 00 Praha, Czech Republic; policht@fld.czu.cz (R.P.); hart@fld.czu.cz (V.H.); 2Forestry and Game Management Research Institute, Strnady 136, 252 02 Jíloviště, Czech Republic; sevcik@vulhm.cz

**Keywords:** *Lyrurus tetrix*, bioacoustics, conservation, acoustic variability

## Abstract

**Simple Summary:**

Geographical variability of acoustic signals is studied more often in songbird species, which does not include the black grouse. The black grouse males use different calls during the mating performance, including vocal rookooing. No studies have yet dealt with a more detailed analysis of these signals, except one. Although the studied hissing calls of this species are classified as non-vocal acoustic signals, individuality was observed in these calls. To discern whether there is also geographical variability across the range or distribution area of the black grouse, we analyzed and compared hissing calls from four countries. Individuals in the Czech Republic, Finland, Scotland, and Russia were recorded during mating seasons. The results of the analysis revealed differences between the four subpopulations, although not at the level of dialect distinction. The study of acoustic variability, supported by genetic research, will help to better understand the links or, conversely, the differentiation processes between subpopulations.

**Abstract:**

The black grouse is a species whose population development requires constant monitoring due to a rapidly declining trend, especially in Central Europe. Variability in the voices of geographically separated populations can aid in counting within individual populations. This has been investigated with the black grouse. However, the variability of the acoustic behavior of black grouse between populations was investigated for the first time. In total, 82 male black grouse were recorded during the lekking season in four countries: the Czech Republic, Scotland, Finland, and Russia. We analyzed recordings of hissing calls, i.e., the non-vocal signal. DFA analysis correctly classified almost 70% of the recordings. The results indicate a certain degree of difference between the grouse populations from the four countries examined. The mean frequency of hissing calls for populations was 1410.71 ± 170.25 Hz, 1473.89 ± 167.59 Hz, 1544.38 ± 167.60 Hz, and 1826.34 ± 319.23 Hz in the Czech Republic, Finland, Russia, and Scotland, respectively. Populations from Scotland and Russia have greater intra-variability compared to grouse from the Czech Republic and Finland, indicating that population density is not the principal factor in the geographical variability of black grouse hissing calls. Range-level differences enhance knowledge and facilitate the assessment of species evolution.

## 1. Introduction

The black grouse (*Lyrurus tetrix*) distribution area extends from Great Britain through Russia to North Korea and from Fennoscandia to the Alps with scattered populations in Central Europe and Albania [1,2]. The populations of this species are experiencing a non-negligible decline, especially in Central and Eastern Europe [3,4,5] and in Fennoscandia [6]. The black grouse as a species is classified as of Least Concern [2]; however, there are countries where it struggles to survive [7,8,9,10,11]. The main reasons for the decline are nest predation and increasing chick and subadult bird mortality [6,12,13,14]. Other factors are changes in the environment [15,16,17,18], parasite infestation [19], predation of adults [20,21,22,23,24,25,26], reduced genetic diversity [10,27], human activities [28,29,30,31,32,33], and global climate changes [6,33,34,35,36]. Based on the above-mentioned reasons, there is a great deal of effort in the above countries to address declining populations [37,38,39,40,41] with the help of reintroductions and effective monitoring methods. Currently, there are several methods for the census of Phasianidae, including black grouse. Baillie [42] presents these census methods for terrestrial breeding bird species: territory mapping, line transects, and point counts. The ideal method for monitoring birds involves a statistically rigorous study design with unbiased estimates against the likely number of observers. Recording males displaying in the spring during the lekking season is the most common method of counting black grouse [43]. Methods currently used in Norway—determination by vocal expressions during display and using dogs on transects—or in combination with physical marking [44,45], are not suitable for low-density populations, such as in Central Europe [19,46,47,48,49]. 

A census based on the recording of the acoustic displays of the male grouse during the courtship on the lek can bring many advantages. It is a non-invasive method that doesn’t disturb birds on the leks and can be more accurate thanks to the individual recognition of males. In black grouse, acoustic signals are divided into two main categories of sounds, which are resonant rookooing and hissing calls [50]. Hissing calls represent a non-vocal signal produced by partial constriction of the windpipe between the lungs and bill [51]. These signals are an important part of courtship on leks because both males and females decide which lek to visit based on acoustic performance [52]. Recently, it was found that the grouse’s non-vocal hissing calls carry an individual specificity that makes it possible to distinguish individuals from each other [53]. 

Therefore, this article is focused on the acoustic expression of the black grouse from a broader perspective. Due to the relatively large area of occurrence of this species [6], there is likely geographical variability in its acoustic display. It could help to differentiate the origin of individuals by their acoustic signals because one of the problems of the black grouse population is its fragmentation and isolation [9,18,27,39,54]. On the other hand, it could theoretically hinder efforts to reintroduce black grouse from remote areas, as during the courtship, the individual’s acoustic display is one of the characteristics to which both females and competing males respond. 

So far, only a few studies have paid attention to this part of the grouse’s behavior. This study aimed to find (i) whether there is an acoustic variability in black grouse male’s display at geographical range and (ii) if it would allow for the determination of the original population of individual black grouse based on their vocal or non-vocal performance.

## 2. Materials and Methods

### 2.1. Study Areas and Recording

The recording of male black grouse took place in four regions: Finland, Russian Federation, and Czech Republic in 2012–2014, and in Scotland in 2019. In total, six locations were visited for recording purposes (Figure 1, Table 1). To prevent the black grouse from being disturbed at its sites, the exact coordinates of the locations where the recording took place are not given. The hissing calls of black grouse were recorded during the spring mating season, which takes place in April and May. All individuals were recorded in the wild. The leks were approached approximately two hours before the arrival of males to ensure an uninterrupted course of data collection. Recording sessions were performed in a portable blind or by hiding in natural vegetation. Each session took about one hour. The distance of the microphone from lekking grouse was 10 m on average. Acoustic signals of male black grouse were recorded with the dictaphone Olympus LP-100—in combination with a Sennheiser ME 66 directional microphone (frequency response 20 Hz–20 kHz ± 2.5 dB) complemented by a K6 powering module. Recordings were saved in .wav format (48 kHz sampling rate, 16-bit sample size). Multiple sites were visited at each of the six locations and, at each site, only individuals that could be distinguished from each other were recorded, usually one or two males per site. Each lek was visited only once to avoid the repeated recording of the same individual. The risk of recording the same individual at two sites was low; according to Borecha, Willebrand, and Nielsen [55], black grouse males show strong fidelity to their leks, and in our study, the leks were at least one kilometer apart. 

### 2.2. Acoustic Analyses

The recordings were analyzed using Raven Pro 1.5 software with a 512 sample size and a Hann window. Only good quality calls with a high signal-to-noise ratio that were non-overlapping with other hissing calls, background noise, and wind were selected for the analysis. The specific area of hissing calls was manually labeled in the analysis to include the beginning to end of the call and the lowest and highest frequencies. Temporal and frequency variables were measured automatically by the software within the indicated areas of the signals. These measurements were entered into the following statistical analysis. The spectrograms were generated in Avisoft-SASLab Pro with FFT length, 1024 sample size, a Hamming window, and 87.5% overlap.

### 2.3. Statistical Analyses

In total, 853 hissing calls from 82 male black grouse from four countries were analyzed. For every individual, a minimum of five calls were included in the analysis (11 ± 5; mean ± SD), and the maximum number of calls was 26. Thirty variables were measured since variables with no or low variance were excluded (Table 2).

To test the potential for individual variation (Potential of Individual Coding—PIC) for each parameter, we compared the inter- and intra-individuality. The PIC ratio was computed for each acoustic parameter by dividing the coefficient of variance between individuals by the mean of the CV intra-values related to each individual [60]. For these tested parameters, a PIC value greater than one means that inter-individual variability is higher than intra-individual variability, and therefore, the monitored variable has the potential to enter further analyses. Significance was tested using the Kruskal–Wallis test. 

The variables were standardized using Z-score transformation (subtracting the mean and dividing by standard deviation). To test individual variations, the stepwise Discrimination Function Analysis (DFA) using IBM SPSS Statistics 24.0 software (IBM Corp., Armonk, USA) was employed. A leave-one-out cross-validation procedure was applied (IBM SPSS Statistics 20) to validate the results of DFA. To evaluate the combined explanatory potential of the DFA variables and for a more appropriate interpretation of the results, the Principal Component Analysis (PCA) was used. 

## 3. Results

### 3.1. Hissing Call Description

The hissing calls of black grouse represent wideband acoustic signals. The energy is spread across a wide frequency range. This type of call can consist of one or two notes; however, the occurrence of a two-syllable form is rare (~*n* < 1%). We excluded two-syllable calls from the analysis. The duration of analyzed calls ranged from 0.1 to 1.21 s (1.00 ± 0.16, mean ± SD).

The Low Frequency ranged from 136.0 to 1411.8 Hz (835.0 ± 171.7 Hz, mean ± SD), and the High Frequency from 1523.8 to 4637.7 Hz (2464.1 ± 435.2 Hz, mean ± SD) for all individuals. The frequency ranged from 775.2 to 3375.0 Hz (1580.3 ± 280.7 Hz, mean ± SD). 

The spectrograms of black grouse recorded in the Czech Republic, Scotland, Russia, and Finland are shown in the figures below (Figure 2). For a representative recording of a hissing call of one individual from each study country, see Appendix A.

### 3.2. Geographical Variation

According to PIC analysis, the variability between countries (inter-variability) was higher than the variability in individuals (intra-variability). All 30 variables could enter the following analyses to differentiate the populations of individual countries based on their vocal activity (Table 3). All Kruskal–Wallis tests were significant (*p* < 0.001). 

From selected parameters, the resulting model included 12 significant acoustic variables (*p* < 0.05; r ≤ 0.89): High Frequency, Aggregate Entropy, Delta Time, Duration 50%, Frequency 5%, Frequency 25%, Frequency 95%, Minimum Entropy, Relative Time 25%, Relative Time 5%, Relative Time 95%, and Time 95% (Table 3). Appendix A shows the values of the measured variables included in the resulting DFA model.

The first discriminant function had Eigenvalues > 1, which explained 70.7% of the variation. The cumulative percentage of explained variance of the first two discriminant functions was 94.2%. The first discrimination function mostly correlated F25% (Frequency 25%) (r = 0.708) and Freq 95% (Frequency 95%) (r = 0.651), and the second discriminant function correlated best with T95% (Time 95%) (r = 0.569) and T5% Rel (Relative Time 5%) (r = 0.419). The Discriminant Function Analysis did not exclude any individual or country. The resulting DFA model correctly classified 68.0% (66.2%, cross-validated result) hissing calls from four countries. The same result from the classification was given by the DFA model with standardized variables (68.0%; 66.2% cross-validated result). Hissing calls from the Czech Republic were correctly classified with 54.5% accuracy (51.7%, cross-validated result), calls from Scotland with 78.2% accuracy (77.1%, cross-validated result), calls from Russia with 57.4% accuracy (56.2%, cross-validated result), and from Finland with 74.0% accuracy (72.0%, cross-validated result). The first two principal components in PCAs captured 54.0% of the variation (Figure 3).

## 4. Discussion

Our results suggest that even a non-vocal signal can carry acoustic variability, and in this case, variability between populations of black grouse from the four countries was studied. The principal component analysis indicates that populations in Scotland and Russia have greater intra-group variability than grouse from the Czech Republic and Finland. Despite this variation, we cannot claim that the four studied black grouse populations have different dialects. 

Studies on the topic of geographical variability in bird acoustic performance have been performed for both song-learning species [61,62,63,64,65,66], and species with innate vocalizations, including orders Gruiformes, Psittaciformes, and Sphenisciformes [67,68,69,70]. Although there have been cases with landfowl that suggest that the evolution of vocal expression in these species may also be affected by encounters [71], it is thought that the black grouse does not belong to the group of song-learning birds. Moreover, the hissing calls that are the subject of this study cannot even be considered song, since they are not produced by a syrinx. All the more interesting is the finding that even non-vocal signals can be characterized by variability at the subpopulation level. Variability in acoustic performance on a population scale may occur for different reasons. One of them is diverse habitats, shaping the vocal signals of birds from different regions [62,69]. These reasons also include long-distance segregation, morphological features, and environmental influences; gender or social selection are also likely to contribute to variability, as found in the four species of Australian fairy-wrens (*Malurus*) [63,64]. In the black grouse distribution areas, there are many fragmented or even isolated populations [18,27,39,54], and as a result of this, acoustic variability can develop between them. 

On the other hand, the opposite can also occur, where different acoustic signal features are displaced by hybridization or competition, or features converge directly to promote the coexistence of individuals [64] while some bird species may not even have a geographical vocal variation [72,73,74,75]. 

So far, the only way to more accurately determine the origin of individuals and distinguish populations is genetic analysis. Although many studies focus on the genetics of the black grouse, information on the genetic structure of populations from different areas is not unified. According to taxonomy, black grouse from all over the European area genetically belong to one species. However, differences may appear at the population level [76]. Populations in Great Britain have some degree of genetic variation, and microsatellites in black grouse show that the population can be divided into at least several management units [77]. An indication of two different genetic groups of the black grouse was discovered in Poland [9]. Eastern Alpine black grouse show similar amounts of genetic variation in populations like Scandinavia [78]. On a general level, studies agree that genetic diversity in the black grouse population is declining [7,10]. The loss of genetic diversity is the result of the decline of the black grouse population and the isolation of subpopulations in individual countries. As our study suggests, the difference between the investigated subpopulations in acoustic performance may play a larger role in the future.

## 5. Conclusions

Our study revealed potential acoustic variability between black grouse populations from four countries across the distribution area. The accuracy of discriminant model classification of hissing calls was 75%, with the highest values for Scotland (79.8%, cross-validated). The black grouse is not classified as a song-learning bird, and that is precisely why it is interesting that acoustic variability at the subpopulation level was detected in this species. One of the reasons for the evolution of variation in bird acoustic performance is the isolation of populations. Monitoring and assessing acoustic variability have the potential to assess population evolution across a distribution range in a non-invasive manner. 

## Figures and Tables

**Figure 1 animals-13-01844-f001:**
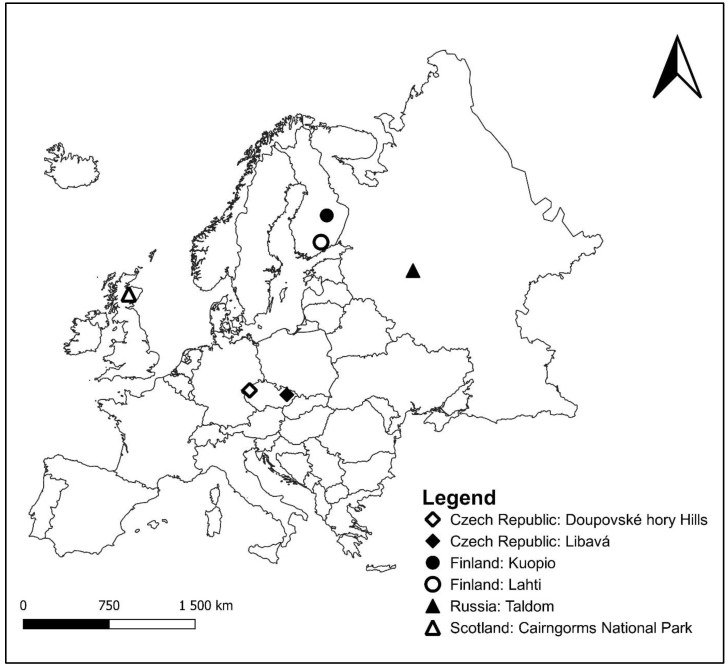
Six locations of black grouse recording: Doupovské hory Hills and Libavá in the Czech Republic, Kuopio and Lahti in Finland, Taldom in Russia, and Cairngorms National Park in Scotland.

**Figure 2 animals-13-01844-f002:**
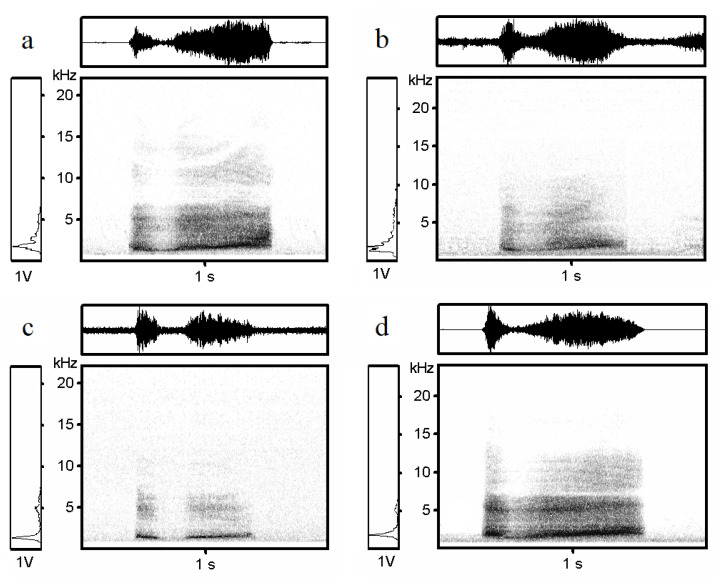
Spectrograms and oscillograms: representative hissing calls of male black grouse from (**a**) Czech Republic, (**b**) Russia, (**c**) Finland, and (**d**) Scotland.

**Figure 3 animals-13-01844-f003:**
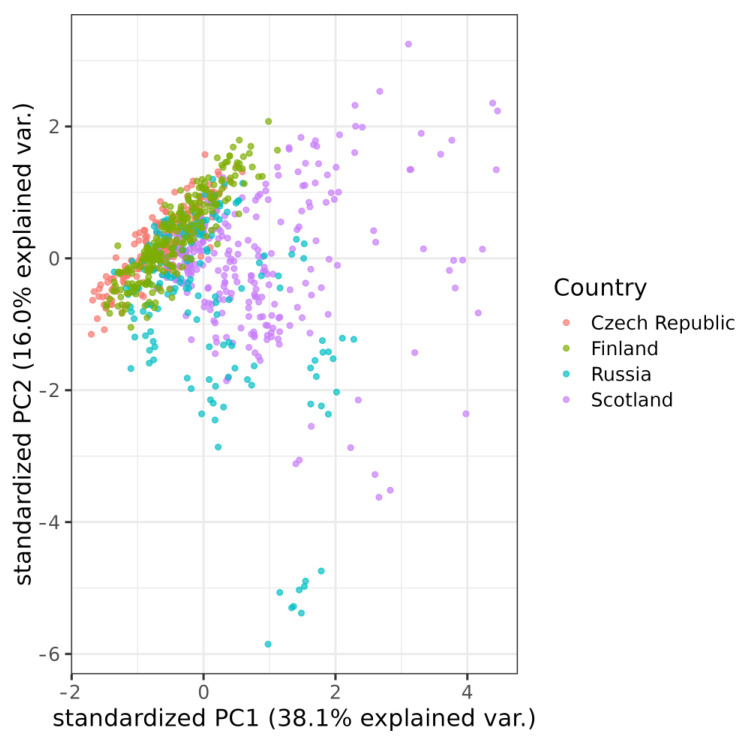
The results of PCA.

**Table 1 animals-13-01844-t001:** Basic information about the black grouse population in four monitored countries.

Country	Estimated Population (Lekking Males)	Location of Study	Males (*n*)	Calls (*n*)	ID of Males
Czech Republic	355	Libavá, Doupovské hory Hills	20	176	1–20
Scotland	3400	Cairngorms National Park	17	262	21–37
Russia	11.3 mil.	Taldom	18	169	38–55
Finland	425,000	Lahti, Kuopio	27	246	56–82
Total			82	853	

Note: Estimated populations refer to dates known to 2022 [56,57,58,59].

**Table 2 animals-13-01844-t002:** Descriptions of acoustic parameters measured in Raven Pro 1.5 that entered statistical analysis.

Acoustic Parameter Name	Abbreviations/Units	Description
Low Frequency	Low Freq (Hz)	The lower frequency bound of the selection.
* High Frequency	High Freq (Hz)	The upper frequency bound of the selection.
* Aggregate Entropy	Agg Entropy (bits)	The aggregate entropy measures the disorder in a sound by analyzing the energy distribution. A pure tone concentrated in only one frequency level corresponds to zero value. Disordered sound that spans more than one frequency level corresponds to higher entropy values. It corresponds to the overall disorder in the sound.
Average Entropy	Avg Entropy (bits)	This entropy is calculated by finding the entropy for each frame in the signal and then taking the average of these values.
Bandwidth 50%	BW 50% (Hz)	The difference between the 25% and 75% frequencies.
Bandwidth 90%	BW 90% (Hz)	The difference between the 5% and 95% frequencies.
Center Frequency	Center Freq (Hz)	The frequency that divides the selection into two frequency intervals of equal energy.
Center Time	Center Time (s)	The point in time at which the selection is divided into two time intervals of equal energy.
Relative Center Time	Center Time Rel,	The point in time at which the selection is divided into two time intervals of equal energy relative to the signal duration.
Delta Frequency	Delta Freq (Hz)	The difference between the upper and lower frequency limits of the selection.
* Delta Time	Delta Time (s)	The difference between Begin Time and End Time for the selection.
* Duration 50%	Dur 50% (s)	The difference between the 25% and 75% times.
Duration 90%	Dur 90% (s)	The difference between the 5% and 95% times.
* Frequency 25%	Freq 25% (Hz)	The frequency that divides the selection into two frequency intervals containing 25% and 75% of the energy in the signal.
* Frequency 5%	Freq 5% (Hz)	The frequency that divides the selection into two frequency intervals containing 5% and 95% of the energy in the signal.
* Frequency 95%	Freq 95% (Hz)	The frequency that divides the selection into two frequency intervals containing 95% and 5% of the energy in the signal.
Length	Length (frames)	The number of frames contained in a selection. For waveform views, the number of frames equals the number of samples in a single channel. For spectrogram and spectrogram slice views, the number of frames equals the number of individual spectra in the selection in one channel. For selection spectrum views, the number of frames always equals 1.
Maximum Entropy	Max Entropy (bits)	Maximum entropy calculated from each frame.
Maximum Frequency	Max Freq (Hz)	The frequency at which Max Power occurs within the selection.
Maximum Time	Max Time (s)	The first time in the selection at which a spectrogram point with power equal to Max Power/Peak Power occurs.
* Minimum Entropy	Min Entropy (bits)	The minimum entropy calculated for a spectrogram slice within the selection bounds.
Relative Peak Time	Peak Time Rel (s)	The first time in the selection at which a sample with amplitude equal to Peak Amplitude occurs.
Time 25%	Time 25% (s)	The time that divides the signal into two time intervals containing 25% and 75% of the energy in the signal.
* Relative Time 25%	Time 25% Rel (s)	The time that divides the signal into two time intervals containing 25% and 75% of the energy in the signal relative to signal duration.
Time 5%	Time 5% (s)	The time that divides the signal into two time intervals containing 5% and 95% of the energy in the signal.
* Relative Time 5%	Time 5% Rel,	The time that divides the signal into two time intervals containing 5% and 95% of the energy in the signal relative to signal duration.
Time 75%	Time 75% (s)	The time that divides the signal into two time intervals containing 75% and 25% of the energy in the signal.
Relative Time 75%	Time 75% Rel,	The time that divides the signal into two time intervals containing 75% and 25% of the energy in the signal relative to signal duration.
* Time 95%	Time 95% (s)	The time that divides the signal into two time intervals containing 95% and 5% of the energy in the signal.
* Relative Time 95%	Time 95% Rel,	The time that divides the signal into two time intervals containing 95% and 5% of the energy in the signal relative to signal duration.

Note: Description of variables measured for hissing calls. Marked variables (*) were included in the final DFA model.

**Table 3 animals-13-01844-t003:** Descriptive statistics and Potential for individual coding.

Variable	DFA	Mean	Min	Max	SE	Mean CVw	CVa	PIC
Low Frequency		834.97	0.00	1411.77	171.74	10.34	20.57	1.99
High Frequency	X	2464.05	1523.80	4637.68	435.49	5.81	17.67	3.04
Agg Entropy	X	3.16	1.89	4.81	0.55	7.31	17.48	2.39
Avg Entropy		2.70	1.86	4.00	0.35	5.15	12.94	2.51
BW 50%		287.38	86.13	1125.00	173.59	31.14	60.41	1.94
BW 90%		794.41	258.40	2156.25	340.44	18.47	42.85	2.32
Center Frequency		1580.29	775.20	3375.00	280.88	4.72	17.77	3.76
Center Time		400.08	0.41	3476.43	571.01	49.42	142.72	2.89
Center Time Relative		0.48	0.06	0.84	0.17	25.75	35.47	1.38
Delta Frequency		1629.08	761.89	3478.26	400.54	9.96	24.59	2.47
Delta Time	X	1.00	0.47	1.52	0.16	9.66	15.57	1.61
Duration 50%	X	0.42	0.04	0.77	0.14	25.83	32.88	1.27
Duration 90%		0.75	0.29	1.18	0.13	11.50	17.62	1.53
Frequency 25%	X	1442.80	187.50	3000.00	239.46	5.73	16.60	2.90
Frequency 5%	X	1222.17	0.00	2250.00	218.52	8.36	17.88	2.14
Frequency 95%	X	2016.58	1291.99	4218.75	426.80	5.80	21.16	3.65
Length		188.50	89.00	467.00	41.32	9.67	21.92	2.27
Maximum Entropy		3.71	2.95	4.83	0.29	3.73	7.92	2.12
Maximum Frequency		1557.99	562.50	3468.75	296.88	7.42	19.06	2.57
Maximum Time		399.91	0.21	3476.39	570.99	49.48	142.78	2.89
Minimum Entropy	X	1.54	0.18	2.83	0.31	10.26	20.07	1.96
Peak Time Relative		0.30	0.01	0.93	0.28	67.33	90.52	1.34
Time 25%		399.85	0.29	3476.30	571.01	49.52	142.81	2.88
Time 25% Relative	X	0.24	0.05	0.66	0.16	43.52	66.79	1.53
Time 5%		399.68	0.17	3475.92	570.99	49.61	142.86	2.88
Time 5% Relative	X	0.08	0.01	0.50	0.05	25.30	61.34	2.42
Time 75%		400.27	0.79	3476.53	571.01	49.33	142.66	2.89
Time 75% Relative		0.67	0.11	0.89	0.10	10.14	15.29	1.51
Time 95%	X	400.43	0.96	3476.66	571.01	49.25	142.60	2.90
Time 95% Relative	X	0.83	0.51	0.97	0.06	4.81	6.96	1.45

Note: Descriptive statistics and Potential for individual coding. (DFA) variables included in the DFA model. (SE) standard error of the mean. (Mean CVw) within individual comparisons. (CVa) between individual comparisons. (PIC) Potential for Individual Coding.

## Data Availability

The data that support the findings of this study will be openly available in any publicly accessible repository, such as Dryad, as soon as this manuscript is accepted.

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
