# Peer review of "Black Grouse Hissing Calls Show Geographic Variability across the Distribution Area"

_animals, 2023, doi:10.3390/ani13111844_

Round 1

Reviewer 1 Report

This is an interesting manuscript addressing an aspect of the behaviour of the Black Grouse that may be of great value in conservation. It is based on a significant amount of field-collected data and the processing and statistical analysis is well explained and well performed. I would like to see an improved introduction, with mention of population status and threats much shortened, so that emphasis is on the subject of the study, i.e., geographical and intra-population variation of hissing calls, not information that is not so relevant to the purpose of the study.

The English language of the manuscript could also be improved. In general, it is readable with, however, a few phrases or sentences that I could not understand. I am not a native English speaker so I cannot make definite suggestions for corrections. However, in the comments below, I have made suggestions wherever I was confident enough.

Specific Comments:

Throughout the text: Change “grouses” to “grouse” (no ‘s’ in the plural)

Throughout the text: According to recent International Ornithological Congress lists (www.worldbirdnames.org), [regrettably!] Tetraonidae is merged with Phasianidae. Thus, you should consider changing Tetraonidae to Phasianidae or to “grouse” or “grouse species”. Also, do not use italics for family names (only genus and species names).

Lines 38 – 58: The first paragraph of the Introduction can be shortened a lot. It is not necessary to mention all the threats faced by the species, nor the reasons for its decline in detail.

Line 38: Replace ‘expands’ by ‘extends’ or ‘stretches’.

Line 45: Correct ‘Fenoscandia’ to double ‘n’

Lines 64 – 70: From “In Norway …” to the end of the paragraph a briefer way can be used to say that some monitoring methods cannot be used in Central Europe due to low population density. For example, “Methods recommended for monitoring in Fennoscandia [44, 45] are not appropriate when population density is low, as in Central Europe”.

Line 76: “… produced by partial constriction located between the lungs and bill [51].”: could it be more correct to say “… by partial constriction of the windpipe between …”?

Line 77: Add ‘an’ before ‘important’ in “These signals are important part…”

Lines 96 – 97: Restructure “Finland, Russian Federation, Scotland, and the Czech Republic, in 2012–2014 and 2019, respectively” to (as appropriate), e.g., “Finland, Russian Federation and Scotland in 2012-2014, and in the Czech Republic in 2019” or “Finland and Russian Federation in 2012-2013, in Scotland in 2014, and the Czech Republic in 2019” or “Finland, Russian Federation, Scotland and the Czech Republic in 2012, 2013, 2014 and 2019 respectively”.

Lines 109 - 110: It is not clear what is meant by “Plural leks in each country were visited, and at every site, only individuals that could be distinguished from each other were recorded, usually one or two males”. Perhaps: “Multiple sites were visited at each [of the six] location[s] and, at each site, only individuals that could be distinguished from each other were recorded, usually one or two males per site. Reading the text, I understand that ‘location’ is a region with a grouse population and lek is synonymous with ‘site’. Is this correct?

Lines 117 – 118: “In every country, at least one location was visited for hissing call recording.” Is not necessary as it adequately explained in the main text.

Table 2, between lines 139 and 140:

a.       I do not understand “… energy only one frequency bin …”. Can it be made clearer?

b.       Correct ‘ght’ in ‘Lenght'

Line 218: Replace ‘syringe’ by ‘syrinx’.

Line 220: Delete “Possible reasons for variability”. It is not necessary, if intended as a headline for the following text, because the discussion is not very long.

Lines 223 – 224: Add a period or a semicolon after ‘influences’ or wherever is appropriate to separate ‘also include’ from ‘are also likely’ (two verbs in one sentence).

Line 244: Use a period after “… individual countries”. Start the next sentence with “As our study suggests, …”

I do not comment on spelling rules, whether American English or other (e.g., for words like behaviour/behavior, centre/center) – it is up to the editor to suggest.

Comments included above. Although English Language could be perfected throughout the text, there are only a few places where the meaning or clarity is seriously affected. These few points should in my opinion be addressed.

Author Response

Dear Reviewer 1

Thank you very much for all your beneficial comments. We are grateful for them, as they have significantly improved our manuscript. In general, we have shortened the manuscript and the language was also checked by native speaker. The particular comments are answered below in detail.

Best regards,

Lucie Hambálková, corresponding author

Specific Comments:

Point 1: Throughout the text: Change “grouses” to “grouse” (no ‘s’ in the plural)

Response: Corrected.

Point 2: Throughout the text: According to recent International Ornithological Congress lists (www.worldbirdnames.org), [regrettably!] Tetraonidae is merged with Phasianidae. Thus, you should consider changing Tetraonidae to Phasianidae or to “grouse” or “grouse species”. Also, do not use italics for family names (only genus and species names).

Response: Thank you very much for this comment, the text was rephrased.

Point 3: Lines 38 – 58: The first paragraph of the Introduction can be shortened a lot. It is not necessary to mention all the threats faced by the species, nor the reasons for its decline in detail.

Response: We agree with your comment. The first paragraph of the Introduction was significantly shortened.

Point 4: Line 38: Replace ‘expands’ by ‘extends’ or ‘stretches’.

Response: Accepted. We replaced ‘expands’ by ‘extends’.

Point 5: Line 45: Correct ‘Fenoscandia’ to double ‘n’

Response: Corrected.

Point 6: Lines 64 – 70: From “In Norway …” to the end of the paragraph a briefer way can be used to say that some monitoring methods cannot be used in Central Europe due to low population density. For example, “Methods recommended for monitoring in Fennoscandia [44, 45] are not appropriate when population density is low, as in Central Europe”.

Response: Accepted. Lines 64 – 70 were shortened.

Point 7: Line 76: “… produced by partial constriction located between the lungs and bill [51].”: could it be more correct to say “… by partial constriction of the windpipe between …”?

Response: Accepted. We improved the line as follows:

„Hissing calls represent a non-vocal signal produced by partial constriction of the windpipe between the lungs and bill [51].”

Point 8: Line 77: Add ‘an’ before ‘important’ in “These signals are important part…”

Response: Corrected.

Point 9: Lines 96 – 97: Restructure “Finland, Russian Federation, Scotland, and the Czech Republic, in 2012–2014 and 2019, respectively” to (as appropriate), e.g., “Finland, Russian Federation and Scotland in 2012-2014, and in the Czech Republic in 2019” or “Finland and Russian Federation in 2012-2013, in Scotland in 2014, and the Czech Republic in 2019” or “Finland, Russian Federation, Scotland and the Czech Republic in 2012, 2013, 2014 and 2019 respectively”.

Response: Accepted. We restructured the lines as follows:

The recording of male black grouse took place in four regions: Finland, Russion Federation and Czech Republic in 2012 – 2014, and in the Scotland in 2019.“

Point 10: Lines 109 - 110: It is not clear what is meant by “Plural leks in each country were visited, and at every site, only individuals that could be distinguished from each other were recorded, usually one or two males”. Perhaps: “Multiple sites were visited at each [of the six] location[s] and, at each site, only individuals that could be distinguished from each other were recorded, usually one or two males per site. Reading the text, I understand that ‘location’ is a region with a grouse population and lek is synonymous with ‘site’. Is this correct?

Response: Yes, the understanding of the location and sites meaning is correct. We improved the lines as suggested:

Multiple sites were visited at each of the six locations and, at each site, only individuals that could be distinguished from each other were recorded, usually one or two males per site.“

Point 11: Lines 117 – 118: “In every country, at least one location was visited for hissing call recording.” Is not necessary as it adequately explained in the main text.

Response: Accepted. We deleted this sentence.

Point 12: Table 2, between lines 139 and 140:

  1. I do not understand “… energy only one frequency bin …”. Can it be made clearer?
  2. Correct ‘ght’ in ‘Lenght'

Response a.: If the energy of the tone is concentrated in only one frequency level, then it is a pure tone and the value of the aggregated entropy corresponds to zero. In the opposite case, when the tone is disordered and spans multiple frequency levels, the aggregate entropy value is greater than zero.

We changed the Description of the acoustic parameter as follows:

The aggregate entropy measures the disorder in a sound by analyzing the energy distribution. A pure tone concentrated in only one frequency level corresponds to zero value. Disordered sound that spans more than one frequency level corresponds to higher entropy values. It corresponds to the overall disorder in the sound.“

Response b.: Corrected.

Point 12: Line 218: Replace ‘syringe’ by ‘syrinx’.

Response: Accepted.

Point 13: Line 220: Delete “Possible reasons for variability”. It is not necessary, if intended as a headline for the following text, because the discussion is not very long.

Response: Accepted.

Point 14: Lines 223 – 224: Add a period or a semicolon after ‘influences’ or wherever is appropriate to separate ‘also include’ from ‘are also likely’ (two verbs in one sentence).

Response: Accepted. We corrected the lines as follows:

“These reasons also include long-distance segregation, morphological features, environ-mental influences; gender or social selection are also likely to contribute to variability, as found in the four species of Australian fairy-wrens (Malurus) [63,64].”

Point 15: Line 244: Use a period after “… individual countries”. Start the next sentence with “As our study suggests, …”

Response: Accepted.

Point 16: I do not comment on spelling rules, whether American English or other (e.g., for words like behaviour/behavior, centre/center) – it is up to the editor to suggest.

Response: We did our best to improve the language. We corrected mistakes including the above mentioned. The text of the manuscript was edited by a native speaker Richard Lee Manore (American English) and, in order not to lose the communicated information in the translation, no major stylistic interventions were made. We have mentioned the native speaker to acknowledge section at the end of article.

Comments on the Quality of English Language

Comments included above. Although English Language could be perfected throughout the text, there are only a few places where the meaning or clarity is seriously affected. These few points should in my opinion be addressed.

Response: We addressed the stated points and did our best to improve the English language.

Reviewer 2 Report

The authors present an interesting study of the geographical variability of a specific hissing call of the Black Grouse. The data were collected at five different sites in four countries, presenting an interesting gradient in the call and its regional variability.

The experimental design, data collection, and analyses are well thought out, and the paper is written well.

However, there are small glitches that the authors must correct. There are several places where the language does not flow and impedes the reading sequence. The authors need to edit the manuscript for language and catch these mistakes. Examples are :

Introduction, Line 44 – “…..classified as a of Least Concern” replace “a” with of

Discussion, Line 218 – “syringe” should read “syrinx”

As above

Author Response

Dear Reviewer 2

Thank you for your comments which improved our manuscript. We did our best to improve the language. In general, we have shortened the manuscript and the language was also checked by native speaker Richard Lee Manore (American English). In order not to lose the communicated information in the translation, no major stylistic interventions were made. The particular comments are answered below in detail.

Best regards,

Lucie Hambálková, corresponding author

Specific Comments:

Point1: Introduction, Line 44 – “…..classified as a of Least Concern” replace “a” with of

Response: Corrected.

Point 2: Discussion, Line 218 – “syringe” should read “syrinx”

Response: Corrected.

Reviewer 3 Report

I think this paper will would be a good addition to the peer reviewed literature. However, whilst I suspect your statistical analysis are appropriate and correct, they are not particularly well described. For example, I think there are three steps to your analysis: 1) PIC (to assess inclusion in DFA); 2) DFA (to identify variables that explain regional differences); and 3) PCA (to evaluate the  combined explanatory potential of the DFA variables), but you do not set this out very clearly. All Kruskal-Wallis test were significant so not strictly needed in table, just mention in text. I think clarity could be improved quite easily by some further editing.

This is comprehensible to me as a native English speaker but wording is not 'natural' in quite a few places (too many to list). Readers for which English is not a strong language may struggle more. However, this could be resolved by a moderate edit.

Author Response

Point 1: I think this paper will would be a good addition to the peer reviewed literature. However, whilst I suspect your statistical analysis are appropriate and correct, they are not particularly well described. For example, I think there are three steps to your analysis: 1) PIC (to assess inclusion in DFA); 2) DFA (to identify variables that explain regional differences); and 3) PCA (to evaluate the combined explanatory potential of the DFA variables), but you do not set this out very clearly. All Kruskal-Wallis test were significant so not strictly needed in table, just mention in text. I think clarity could be improved quite easily by some further editing.

Dear reviewer,

Thank you for your comments. The Materials and Methods section has been edited. As correctly understood, 3 steps of statistical analysis were used. We explained the use of PCA as it was not previously stated. Information about Kruskal-Wallis test signification was moved to the main text.

Point 2: This is comprehensible to me as a native English speaker but wording is not 'natural' in quite a few places (too many to list). Readers for which English is not a strong language may struggle more. However, this could be resolved by a moderate edit.

Response: We understand that the wording is perceived as not 'natural', however, the text of the manuscript was edited by a native speaker Richard Lee Manore (American English) and, in order not to lose the communicated information in the translation, no major stylistic interventions were made.

Best regards,

Lucie Hambálková, corresponding author